# Effect of Different Approaches to Antimicrobial Therapy with Cefmetazole and Meropenem on the Time to Defervescence in Non-Severe Extended-Spectrum β-Lactamase-Producing *Escherichia coli* Bacteremia

**Takanobu Hoshi** [1], **Satoshi Fujii** [2,*], **Kei Watanabe** [3], **Yuta Fukumura** [3], **Koji Miyazaki** [3], **Madoka Takahashi** [3], **Sakae Taniguchi** [3], **Shingo Kimura** [3], **Arisa Saito** [3], **Naoki Wada** [4], **Masaji Saijo** [5], **Kazunori Yamada** [6], **Kuninori Iwayama** [1], **Marie Itaya** [3] and **Hideki Sato** [1]

1    Division of Clinical Pharmacy, Department of Pharmacy, Faculty of Pharmaceutical Sciences, Hokkaido University of Science, Sapporo-shi 006-8585, Hokkaido, Japan; hoshi-t@hus.ac.jp (T.H.); iwayama-k@hus.ac.jp (K.I.); h.satoh@hus.ac.jp (H.S.)
2    Department of Hospital Pharmacy, Sapporo Medical University Hospital, Sapporo-shi 060-8543, Hokkaido, Japan
3    Department of Pharmacy, Sapporo Tokushukai Hospital, Sapporo-shi 004-0041, Hokkaido, Japan; nowarist.k1@gmail.com (K.W.); fukumura.yuuta@gmail.com (Y.F.); miyakoji.p0219@gmail.com (K.M.); cubby.927@gmail.com (M.T.); pharm.taniguchis@gmail.com (S.T.); shingo.kimura1995@gmail.com (S.K.); atsuuu22@gmail.com (A.S.); yakkyoku18@satutoku.jp (M.I.)
4    Department of Clinical Laboratory, Sapporo Tokushukai Hospital, Sapporo-shi 004-0041, Hokkaido, Japan; nawada@satutoku.jp
5    Department of Primary Care, Sapporo Tokushukai Hospital, Sapporo-shi 004-0041, Hokkaido, Japan; kyokui070014@gmail.com
6    Department of Pharmacy, Nakamura Memorial Hospital, Sapporo-shi 060-8570, Hokkaido, Japan; kazuyama0813@gmail.com
*    Correspondence: fujii.satoshi@sapmed.ac.jp; Tel.: +81-11-611-2111

**Abstract:** Carbapenems are antimicrobial agents commonly used to treat extended-spectrum β-lactamase (ESBL)-producing bacteria. Although cefmetazole (CMZ) is considered effective for ESBL-producing *Escherichia coli* (ESBL-EC) bacteremia, previous studies showed its limitations, including the influence of the initial antimicrobial agent. Here, we examined the effects of different approaches to antimicrobial therapy with CMZ and meropenem (MEPM) on the time to defervescence in ESBL-EC bacteremia. Notably, the influence of previous antimicrobial agents was excluded. Inpatients with ESBL-EC detected in blood cultures between April 2018 and March 2023 were included and assigned to CMZ (*n* = 14), MEPM (*n* = 8), de-escalation to CMZ (dCMZ; *n* = 9), or escalation to MEPM (eMEPM; *n* = 11) groups. The median time to defervescence was 3.5, 1.0, 2.0, and 4.0 days in the CMZ, MEPM, dCMZ, and eMEPM groups, respectively, with no significant differences. Cox proportional hazards analysis showed a significant difference in the hazard ratio (95% confidence interval) of 0.378 (0.145–0.984) for the time to defervescence with CMZ versus MEPM (*p* = 0.046). The extent of a delayed time to defervescence is greater with early CMZ administration than with MEPM administration in patients with non-severe ESBL-EC bacteremia.

**Keywords:** cefmetazole; meropenem; extended-spectrum β-lactamase; *Escherichia coli* bacteremia

## 1. Introduction

Extended-spectrum β-lactamases (ESBLs) are a type of β-lactamase enzyme with an extended substrate due to genetic mutations. Among the various types of resistance genes, such as TEM, SHV, and CTX-M, CTX-M is the most prevalent worldwide [1,2]. ESBL-producing *Enterobacterales* (ESBL-PE) are widespread in the community and have been reported in the intestinal tracts of healthy individuals with no history of antimicrobial

therapy [3]. The incidence of *Escherichia coli*-producing ESBL (ESBL-EC) is increasing [4]. Individuals with ESBL-EC bloodstream infections (BSI) are reported to have a 14% longer hospital stay, a higher risk of recurrence [4], and higher mortality in critically ill patients in the intensive care unit [5] than those with non-ESBL-EC BSI. The origin enzyme of the CTX-M type is present on the chromosome of *Kluyvera* spp. [6,7] and is characterized by the specific degradation of cefotaxime; however, cephamycin-based drugs are often effective against ESBLs, including CTX-M. Carbapenem (CBP) is the first-line treatment for ESBL-PE. Even the isolation of ESBL-EC from urine samples has been reported to affect the use of CBP [8]. Recently, carbapenemase-producing and carbapenem-resistant *Enterobacterales* have emerged worldwide owing to the increased inappropriate use of antimicrobial agents [9]. Consequently, there is a pressing need for CBP-independent treatment strategies to address ESBL-PE infections [10]. To date, previous studies have reported the non-inferiority of clinical outcomes and 30-day mortality of CBP and non-CBP (tazobactam/piperacillin [TAZ/PIPC] and cefmetazole [CMZ]) for bacteriuria [11,12]. Furthermore, in normal immune adults, non-inferiority of 30-day mortality between CBP and non-CBP (TAZ/PIPC and CMZ) [13] as well as the efficacy of CMZ as an empirical treatment of *Escherichia coli* (ESBL-EC) bacteremia has been reported [14]. However, previous studies have included the effect of initial antimicrobials, thereby limiting the assessment for the clinical efficacy of non-CBPs. Therefore, developing evidence of non-CBP administration from the beginning of ESBL-PE bacteremia to the end of treatment is important. Nevertheless, facilities using the disk diffusion method of measuring ESBLs by adding clavulanic acid often experience difficulty collecting cases of non-CBPs administered early during the treatment because obtaining ESBL determination results requires several days.

The treatment of ESBL-EC bacteremia with early CMZ administration has long been recommended by Sapporo Tokushukai Hospital. In April 2022, we also introduced the Bio Fire blood culture identification (BCID2) FilmArray® Torch System (FilmArray®, Sanwa Medical, Inc, Matsuyama, Japan) to detect ESBLs and administered CMZ earlier. Therefore, this study aimed to compare the differences in clinical efficacy between early CMZ and meropenem (MEPM) administration in ESBL-EC bacteremia and to determine the usefulness of CMZ administration. The differences in clinical efficacy in cases of de-escalation from broad-spectrum antimicrobial agents to CMZ and escalation from CMZ to MEPM were also investigated to compare the impact of different approaches to antimicrobial therapy on clinical efficacy.

## 2. Materials and Methods

### 2.1. Study Design and Participants

In this retrospective study, we analyzed data of patients positive for ESBL-ECs from blood cultures between April 2018 and March 2023 in Sapporo Tokushukai Hospital. The patients were assigned to the CMZ, MEPM, CMZ de-escalation (dCMZ), or MEPM escalation (eMEPM) groups. The CMZ group included patients who were started with CMZ by the third day after the positive blood culture date and continued receiving treatment with CMZ. The MEPM group included those who were started with MEPM by the third day after the positive blood culture date and completed treatment with MEPM. The CMZ and MEPM groups included patients who had received antimicrobial agents that were ineffective against ESBL-EC prior to CMZ and MEPM administration. The dCMZ group included patients who received MEPM by the third day after the positive blood culture date and were de-escalated to CMZ after the third day. The eMEPM group included those who received effective antimicrobial therapy for ESBL-EC as initial therapy and were escalated to MEPM after day 3.

Exclusion criteria were as follows: patients receiving a combination of an antimicrobial agent, no antimicrobial administration, transfer, dialysis, critical illness (the administration of noradrenaline, vasopressin, dopamine, dobutamine, or intensive care), positive blood culture for multiple organisms, and patients who had received a susceptible antimicrobial

agent at least 2 days before the positive blood culture date. Patients who did not fall into the category of critical illness were labelled as "non-severe".

*2.2. Study Endpoints and Data Collection*

The primary endpoint was the time to defervescence. The days when the maximum body temperature was 37.5 °C or lower for three consecutive days were defined as days of resolution of fever, and the time from the blood culture-positive day to fever resolution was calculated as the duration of defervescence. Secondary endpoints included patient background (age, sex, respiratory rate, systolic blood pressure, comorbidities, source of bacteremia), laboratory values (C-reactive protein, white blood cell, albumin, platelet, prothrombin activity, total-bilirubin, estimated glomerular filtration rate [15], aspartate aminotransferase, alanine aminotransferase, and blood sugar), duration of treatment (days), length of hospital stay (days), antipyretic agents administered or not, antimicrobial costs (USD), 30-day mortality rate (%), and Minimum Inhibitory Concentration (MIC) of CMZ for isolated bacteria.

For blood culture, blood was collected in blood fluid culture resin bottles (22F and 23F) (Becton Dickinson: Japan BD, Tokyo, Japan) and cultured for 7 days at 35 °C maximum using the BACTEC FX system (Japan BD). Identification of bacteria before the introduction of FilmArray® was performed using a matrix-assisted laser desorption/ionization time of flight mass spectrometry biotyper (Bruker Daltonics, Kanagawa, Japan). Drug susceptibility testing was determined according to Clinical and Laboratory Standards Institute (CLSI) M100-S26 on the Micros can Negative MIC panel for Enterobacterales 2J panel using the Micros can WalkAway 40 Plus (Beckman Coulter, Inc, Tokyo, Japan) [16]. Furthermore, the ESBL phenotype confirmation test was performed according to CLSI criteria [16]. After introduction of FilmArray® (Sanwa Medical, Inc, Matsuyama, Japan), Gram-negative bacteria with CTX-M gene detected were considered positive for ESBL using BCID2.

*2.3. Statistical Analysis*

Continuous variables were compared using the Kruskal–Wallis test, and the results are presented as median (interquartile range). Categorical variables were compared using Fisher's exact test. Cox proportional hazards analysis was conducted, incorporating the time to defervescence, treatment duration, and hospitalization as explanatory variables. A *p*-value < 0.05 was considered statistically significant. Statistical analysis was performed using JMP® Pro 17.0.0. software (SAS Institute Inc., Cary, NC, USA).

**3. Results**

*3.1. Target Patients*

The flow chart of target patients is shown in Figure 1. Of the 120 patients screened during the study period, 78 were excluded, and 42 were finally included in the study. Twenty-two patients were given an antimicrobial agent effective against ESBL-EC within 3 days of the bacteremia and completed treatment with that agent alone. Further, 14 and 8 patients were treated with CMZ and MEPM, respectively. A total of 20 patients, including 9 treated with dCMZ and 11 with eMEPM, were switched to other effective antimicrobial agents (de-escalation or escalation) after starting an effective antimicrobial for ESBL-EC.

Of the 120 patients screened during the study period, 78 were excluded, and 42 were finally included in the study. Depending on whether the initial treatment was continued, eligible patients were divided into the CMZ group (*n* = 14), MEPM group (*n* = 8), dCMZ group (*n* = 9), and eMEPM group (*n* = 11).

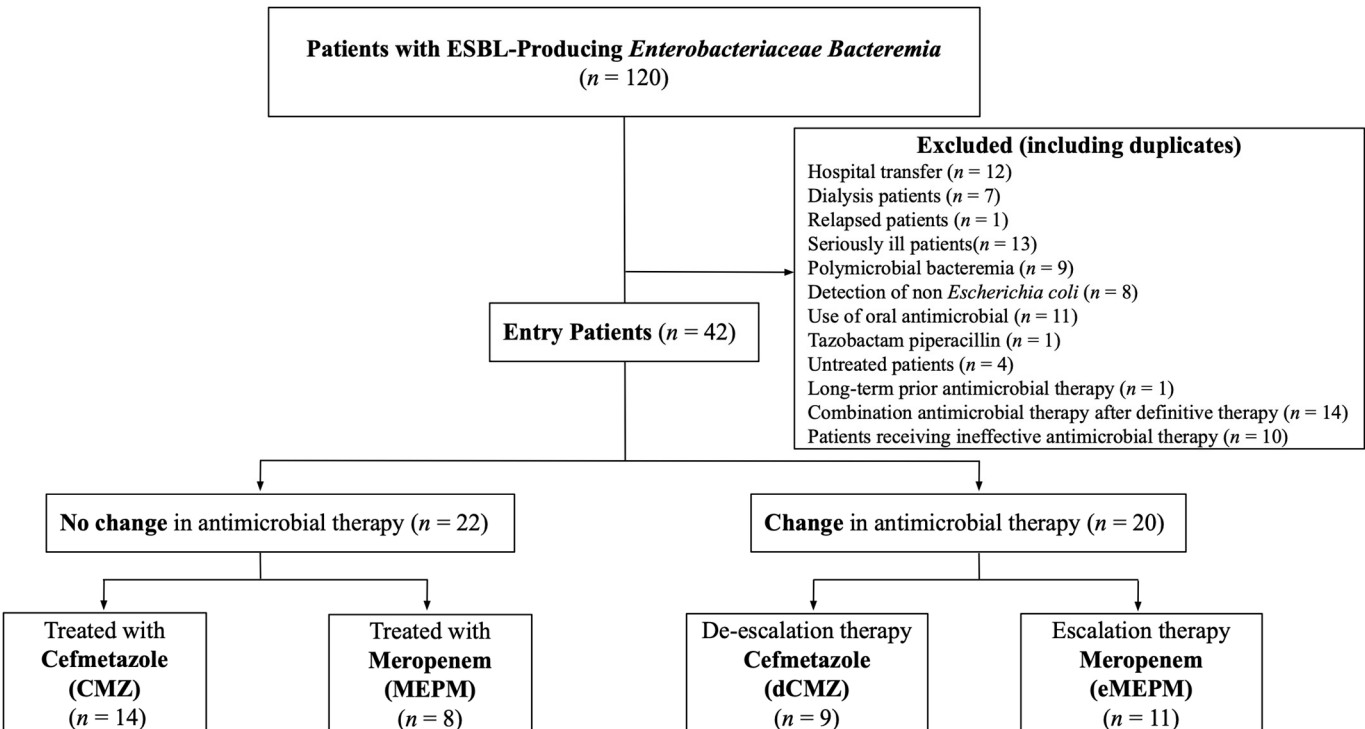

**Figure 1.** Flow diagram of the patient selection process.

*3.2. Primary Endpoint*

The median time to defervescence was 3.5, 1.0, 2.0, and 4.0 days in the CMZ, MEPM, dCMZ, and eMEPM groups, respectively. Notably, no significant differences among the groups were observed (Table 1). The percentage of antipyretics used in the CMZ, MEPM, dCMZ, and eMEPM groups was 21.4%, 12.5%, 0%, and 18.2%, respectively, with no significant differences (Table 1).

**Table 1.** Comparison of patient background and clinical outcomes between CMZ and MEPM groups.

| Characteristic | CMZ (n = 14) | MEPM (n = 8) | dCMZ (n = 9) | eMEPM (n = 11) | p-Value |
|---|---|---|---|---|---|
| Age (years) | 87.5 (81.8–91.3) | 86.5 (82.0–89.8) | 87.0 (79.5–89.0) | 85.0 (81.0–87.0) | 0.849 |
| Sex, male (%) | 4 (28.7) | 3 (37.5) | 5 (55.6) | 7 (63.6) | 0.302 |
| Respiratory rate | 18 (17–20) | 17 (15–22) | 18 (17–24) | 20 (18–25) | 0.311 |
| SBP (mmHg) | 136 (118–146) | 107 (106–136) | 125 (97–147) | 125 (106–155) | 0.464 |
| **Comorbidities (%)** | | | | | |
| Renal dysfunction (eGFR < 30 mL/min/1.73 m$^2$) | 3 (21.4) | 2 (25.0) | 2 (22.2) | 2 (18.2) | 0.988 |
| Cerebrovascular disease | 6 (42.9) | 3 (37.5) | 2 (22.2) | 4 (36.4) | 0.792 |
| Circulatory disease | 6 (42.9) | 2 (25.0) | 2 (22.2) | 2 (18.2) | 0.530 |
| Diabetes | 5 (35.7) | 2 (25.0) | 0 (0) | 3 (27.3) | 0.263 |
| Cancer | 5 (35.7) | 3 (37.5) | 4 (44.4) | 2 (18.2) | 0.629 |
| **Source of bacteremia (%)** | | | | | 0.329 |
| Urinary tract | 10 (71.4) | 7 (87.5) | 8 (88.9) | 9 (81.8) | |
| Respiratory | 1 (7.1) | 1 (12.5) | 0 (0) | 2 (18.2) | |
| Biliary tract | 0 (0) | 0 (0) | 0 (0) | 0 (0) | |
| Others | 3 (21.4) | 0 (0) | 1 (11.1) | 0 (0) | |

**Table 1.** *Cont.*

| Characteristic | CMZ (n = 14) | MEPM (n = 8) | dCMZ (n = 9) | eMEPM (n = 11) | p-Value |
|---|---|---|---|---|---|
| Laboratory Data | | | | | |
| CRP (mg/dL) | 6.9 (3.9–11.9) | 9.7 (3.5–13.3) | 10.6 (5.9–16.9) | 4.5 (0.7–15.3) | 0.521 |
| WBC ($10^3$/μL) | 11.5 (8.3–16.3) | 14.7 (7.1–25.3) | 13.4 (11.0–14.4) | 12.5 (9.2–20.3) | 0.916 |
| Albumin (mg/dL) | 2.4 (2.1–3.2) | 2.5 (2.2–3.1) | 2.8 (2.3–3.1) | 3.4 (3.0–3.6) | 0.040 |
| Platelet ($\times 10^4$/μL) | 14.8 (11.9–17.4) | 15.6 (11.6–25.2) | 17.5 (9.4–24.2) | 15.3 (11.7–25.4) | 0.957 |
| Prothrombin activity (%) | 79.3 (66.9–88.2) | 76.4 (25.2–93.5) | 66.6 (65.5–72.9) | 73.3 (57.0–85.1) | 0.473 |
| Total-bilirubin (mg/dL) | 0.8 (0.3–1.1) | 0.8 (0.5–1.2) | 0.7 (0.5–1.4) | 1.1 (0.4–1.3) | 0.867 |
| eGFR (mL/min/1.73 $m^2$) | 42.7 (32.9–89.3) | 56.5 (32.9–92.3) | 58.4 (31.1–74.0) | 46.8 (39.8–61.2) | 0.908 |
| AST (mg/dL) | 29 (18–75) | 23 (15–28) | 20 (15–44) | 21 (18–37) | 0.556 |
| ALT (mg/dL) | 17 (10–77) | 20 (12–25) | 17 (10–27) | 16 (10–21) | 0.859 |
| Blood sugar level (mg/dL) | 123 (107–168) | 111 (94–132) | 90.5 (69–118) | 122 (96–147) | 0.125 |
| CMZ MIC distribution (%) | | | | | 0.090 |
| MIC < 4 | 13 (92.9) | 7 (87.5) | 9 (100.0) | 7 (63.6) | |
| MIC < 8 | 1 (7.1) | 1 (12.5) | 0 (0) | 4 (36.4) | |
| Others | | | | | |
| Time to defervescence (day) | 3.5 (1.0–7.3) | 1.0 (1.0–3.8) | 2.0 (1.0–3.0) | 4.0 (3.0–7.0) | 0.069 |
| Use of antipyretics for more than 3 days (%) | 3 (21.4) | 1 (12.5) | 0 (0) | 2 (18.2) | 0.524 |
| Duration of treatment initiation (day) | 0 (−1–1) | 0 (−2–0.5) | 0 (0–0.5) | 0 (0–0) | 0.267 |
| Duration of treatment (day) | 14.5 (12.8–17.0) | 13.0 (12.0–15.0) | 13.0 (12.3–19.3) | 17.0 (15.0–19.0) | 0.069 |
| Duration of hospitalization (day) | 24.0 (14.0–32.3) | 23.0 (12.0–47.0) | 37.0 (24.0–55.3) | 22.0 (19.0–27.0) | 0.232 |
| Readmission Rate (%) | 1 (7.1) | 0 (0) | 0 (0) | 0 (0) | 0.562 |
| Cost of antimicrobial therapy (USD) | 98.2 (75.1–132.4) | 216.9 (143.5–243.6) | 167.4 (138.1–191.7) | 236.2 (181.0–265.1) | <0.001 |
| 30-day recurrence rate (%) | 0 (0) | 0 (0) | 0 (0) | 1 (9.1) | 0.409 |
| 30-day mortality rate (%) | 0 (0) | 1 (12.5) | 1 (11.1) | 0 (0) | 0.376 |

Data are presented as median (interquartile range [IQR]: 25th to 75th percentile) and numbers (with percentages). $p < 0.05$ was considered statistically significant. MIC was measured by the micro-liquid dilution method. The quantitative data were compared using the Kruskal–Wallis test, Nominal scales were compared using Fisher's exact test. Abbreviations: SBP, Systolic blood pressure; CRP, C-reactive protein; eGFR, estimated glomerular filtration rate; AST, aspartate aminotransferase; ALT, alanine aminotransferase; WBC, white blood cell; CMZ, cefmetazole; MEPM; meropenem; MIC, minimum inhibitory concentration. dCMZ: CMZ de-escalation group. eMEPM: MEPM escalation group.

### 3.3. Secondary Endpoints

The results of the secondary endpoints are presented in Table 1. Significant differences in antimicrobial costs were observed ($p < 0.001$). The time to mortality for the two deaths, which was not included in the calculation of treatment duration, was 19 and 28 days, respectively. For MIC distribution, there were no strains showing MICs other than <4 and <8. Both <4 and <8 are susceptible by CLSI criteria, but the percentage of <8 in the CMZ, MEPM, dCMZ, and eMEPM groups were 7.1%, 12.5%, 0%, and 36.4%, respectively. Regarding the source of bacteremia, the urinary tract was the source in 71.4% (10/14) of patients in the CMZ group, 87.5% (7/8) in the MEPM group, 88.9% (8/9) in the dCMZ group, and 81.8% (9/11) in the eMEPM group.

*3.4. Cox Proportional Hazards Analysis for the Time to Defervescence*

The Cox proportional hazards analysis, with time to defervescence and treatment and hospitalization durations as outcome variables, revealed significant differences. The hazard ratio (HR) (95% confidence interval [CI]) for the time to defervescence between CMZ and MEPM was 0.378 (0.145–0.984), indicating a significant difference ($p = 0.046$) (Table 2). Similarly, a significant difference in treatment duration, with an HR of 0.276 (0.098–0.774), was observed for eMEPM compared with for MEPM ($p = 0.015$). However, no significant difference was observed in hospitalization duration between the two groups. To account for various sources of bacteremia, we excluded "other" cases within the CMZ group and conducted an identical Cox proportional hazards model analysis. The objective variables including time to defervescence and treatment and hospitalization durations showed significant differences, with a HR (95% CI) of 0.312 (0.112–0.872) for the time to defervescence between CMZ and MEPM ($p = 0.026$). Furthermore, a significant difference in the treatment duration with an HR of 0.286 (0.102–0.803) for eMEPM relative to MEPM ($p = 0.017$) was observed. However, no significant difference in hospitalization duration was observed between the groups.

**Table 2.** Cox proportional hazards model analysis for the time to defervescence, duration of treatment, and duration of hospitalization.

|  | Hazard Ratio | *p*-Value | 95% Confidence Interval |
|---|---|---|---|
| Time to defervescence (day) |  |  |  |
| CMZ (vs. MEPM) | 0.378 | 0.046 | 0.145–0.984 |
| dCMZ (vs. MEPM) | 0.922 | 0.868 | 0.352–2.412 |
| eMEPM (vs. MEPM) | 0.389 | 0.056 | 0.147–1.022 |
| Duration of treatment (day) |  |  |  |
| CMZ (vs. MEPM) | 0.476 | 0.127 | 0.184–1.235 |
| dCMZ (vs. MEPM) | 0.328 | 0.053 | 0.106–1.013 |
| eMEPM (vs. MEPM) | 0.276 | 0.015 | 0.098–0.774 |
| Duration of hospitalization (day) |  |  |  |
| CMZ (vs. MEPM) | 0.945 | 0.906 | 0.373–2.397 |
| dCMZ (vs. MEPM) | 0.583 | 0.303 | 0.209–1.625 |
| eMEPM (vs. MEPM) | 1.067 | 0.911 | 0.403–2.773 |

Abbreviations: CMZ, cefmetazole; MEPM, meropenem; vs, versus. dCMZ: CMZ de-escalation group, and eMEPM: MEPM escalation group.

## 4. Discussion

This study aimed to investigate the impact of different approaches to antimicrobial therapy for ESBL-EC bacteremia on clinical efficacy, with time to defervescence as the primary endpoint for clinical efficacy. We found a greater delayed time to defervescence with early CMZ administration than with MEPM administration in patients with non-severe ESBL-EC bacteremia. Notably, previous studies used 30-day mortality to evaluate CMZ efficacy and reported no significant difference between CMZ and CBP administration [14,17]. In addition, previous studies have shown that the duration of treatment for ESBL-EC bacteremia ranges approximately from 8 to 12 days. Hence, for an accurate calculation of the 30-day mortality, patients who remain hospitalized after treatment should be included, but previous studies have not mentioned this point. Given the challenges in accurately assessing survival at 30 days, we chose not to use 30-day mortality as the primary endpoint. To the best of our knowledge, no reports have used the time to defervescence as a measure of clinical efficacy in ESBL-EC bacteremia. Assessing the differences in the time to defervescence resulting from different approaches to antimicrobial therapy will provide valuable insights for the development of antimicrobial treatment strategies.

Fukuchi et al. [17] demonstrated the efficacy of CMZ administration in patients with a stable clinical status of ESBL-PE bacteremia. Mita et al. [18] investigated mortality risk factors for ESBL-PE bacteremia and reported ICU admission, shock vitals, and other risk

factors for mortality. This evidence for CMZ administration in ESBL-PE bacteremia is mostly based on mild cases, and there is a paucity of evidence examining the efficacy of CMZ in severe cases [19]. In the present study, we also examined the efficacy of CMZ by excluding severe cases. In univariate analysis of the primary endpoint, no significant difference in the time to defervescence was observed between the MEPM and CMZ groups. However, the median time to defervescence in the CMZ group was longer than that in the MEPM group, suggesting that the time to defervescence may be prolonged. In contrast, the secondary endpoint of 30-day mortality was similar among the groups, similar to previous studies (Table 1). Further, Cox proportional hazards analysis showed an HR (95% CI) of 0.378 (0.145–0.984) for the time to defervescence with CMZ versus MEPM, indicating a more extended time to defervescence in the CMZ group than in the MEPM group. This is an important finding when initially selecting CMZ. Based on these findings, we suggest that differences in the time to defervescence should be considered when selecting the initial antimicrobial agent (CMZ or MEPM) for ESBL-EC bacteremia.

Notably, a certain number of treatment-refractory cases (eMEPM) exist among early CMZ-treated patients. Hamada et al. [20] reported 1 g q8 h as the appropriate dose of CMZ for invasive urinary tract infections in ESBL-EC with MIC $\leq 4$. Takemura et al. [21] calculated target PK/PD parameter values using a neutropenic femoral mouse model and showed that clinically approved CMZ dosage regimens of 1.0 g (30-min intravenous infusion) q8 h for MIC $\leq 0.125$ μg/mL and 1.0 g q6 h for MIC $\leq 0.25$ μg/mL achieved the target PK/PD parameter. Among the two groups (CMZ and eMEPM) that received early administration of CMZ, the percentage of MIC < 8 was 36.4% (4/11) for eMEPM and 7.1% (1/14) for CMZ, which was not significantly different; however, the percentage was higher for eMEPM. This suggests that differences in isolated MICs may affect the clinical efficacy of CMZ. Therefore, it may be necessary to select antimicrobial agents based on the MIC isolation of ESBL-EC.

In this study, we were not able to clarify the factors that contributed to the ineffectiveness of early CMZ administration. In the future, it will be essential to clarify the criteria for early CMZ administration in ESBL-EC bacteremia, including differences in MICs, and identify factors that affect clinical efficacy. Furthermore, we consider that there were issues related to the genotype of ESBL-EC in evaluating the clinical outcome of antimicrobial therapy in this study. Notably, we did not identify the ESBL genes before the introduction of FilmArray®, which may result in the inclusion of genotypes other than CTX-M. Unfortunately, we could not exclude the impact of these factors on clinical response. Nakai et al. [22] reported that ESBL-EC accounted for 80% of isolates and CTX-M-9 was the most common in an epidemiological study of fecal carriage among older individuals in Japan. Therefore, considering the high prevalence of the CTX-M type in Japanese epidemiology, we believe that the presence or absence of ESBL gene identification during this study period had a slight impact on clinical efficacy. However, the usefulness of the early administration of CMZ in areas where resistance genes other than the CTX-M genotype are frequently isolated has not been studied. Therefore, careful decision-making is needed in the selection of antimicrobial agents for ESBL-EC bacteremia in areas where these genotypes are frequently isolated.

This study had several limitations. First, non-inferiority studies to evaluate the equivalence between MEPM and CMZ are lacking. Increasing the evidence from backward-looking studies is desirable, as prospective studies are difficult to conduct. Second, the number of patients in this single-center study was low, and a multicenter collaborative study should be conducted for wider applicability. Third, we could not rule out factors other than antipyretic (e.g., diversity of host immune response, drug fever, and non-infectious fever) as influencing the time to defervescence. However, the results obtained in this study are a rare finding that examines the efficacy of early CMZ administration by excluding the influence of initial antimicrobial agents while using the time to defervescence as an indicator. We hope that our findings contribute toward the establishment of an antimicrobial strategy against ESBLs-EC bacteremia.

## 5. Conclusions

The extent of a delayed time to defervescence is greater with early CMZ administration than with MEPM administration in patients with non-severe ESBL-EC bacteremia. Therefore, differences in the time to defervescence should be critically considered upon selecting the initial antimicrobial agent to treat ESBL-EC bacteremia.

**Author Contributions:** Conceptualization, T.H. and S.F.; methodology, T.H. and K.I.; software, T.H.; validation, T.H., S.F. and H.S.; formal analysis, T.H.; investigation, T.H., K.W., Y.F., K.M., M.T., S.T., A.S., S.K. and N.W.; resources, T.H.; data curation, T.H. and M.I.; writing—original draft preparation, T.H.; writing—review & editing, T.H., M.S., S.F., K.Y. and H.S.; visualization, T.H.; supervision, S.F., K.Y. and H.S.; project administration, T.H. All authors have read and agreed to the published version of the manuscript.

**Funding:** This research received no external funding.

**Institutional Review Board Statement:** This study was conducted in accordance with the Declaration of Helsinki and approved by the Institutional Review Board of the Mirai Medical Research Center Inc. (Tokyo, Japan) (Ethical Review Number.: TGEO2153-010, approved on 25 December 2023).

**Informed Consent Statement:** Informed consent was obtained for an opt-out.

**Data Availability Statement:** The data presented in this study are available on request from the corresponding author, e.g., privacy or ethical. The data are not publicly available due to private health information restrictions within the Sapporo Tokushukai Hospital system.

**Acknowledgments:** We thank the Antimicrobial Stewardship Team for contributing to the daily support of infectious disease management.

**Conflicts of Interest:** The authors declare no conflicts of interest.

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
