# Peer review of "Effect of Different Approaches to Antimicrobial Therapy with Cefmetazole and Meropenem on the Time to Defervescence in Non-Severe Extended-Spectrum β-Lactamase-Producing Escherichia coli Bacteremia"

_2036-7449, doi:10.3390/idr16010003_

Round 1

Reviewer 1 Report (New Reviewer)

Comments and Suggestions for Authors

Dear Authors, 

thank you for the opportunity to review the paper entitled "Effect of different approaches to antimicrobial therapy with cefmetazole and meropenem on the time to deferverscence in non-severe extended-spectrum beta-lactamase-producing Escherichia coli bacteremia" 

It is a very interesting paper, and indeed, the tool that you choose "time to defervescence" as a measure of clinical efficacy of antibacterial treatment in ESBL-CE bacteremia is quite unique but also very clear when evaluate. 

I have some minor comments or questions, stated below: 

- line 94 - which is the point of this phrase? ("Patients who did not (...) were labelled as "non-severe") - they were excluded or not?! I did not notice this term to be used later on. 

- table 1 - Source of bacteremia - did you have a patient with more than one source of bacteremia?! In group dCMZ (n=9), there are presented 10 sources (8 UTI, 1 respiratory and 1 biliary) - is it so?

- line 211 - ESBL-PE - is not explained (PE)

Author Response

Dear Reviewer 1,

I appreciate your valuable input.

Here is my response.

・line 94 - which is the point of this phrase? ("Patients who did not (...) were labelled as "non-severe") - they were excluded or not?! I did not notice this term to be used later on.

This study includes patients with non-severe ESBL-EC.

Therefore, the definition of nonsevere disease is given here.

・table 1 - Source of bacteremia - did you have a patient with more than one source of bacteremia?! In group dCMZ (n=9), there are presented 10 sources (8 UTI, 1 respiratory and 1 biliary) - is it so?

There was a mistake in the description. Thank you for pointing this out.

The correct numbers are 8 UTI and 1 Others.

・line 211 - ESBL-PE - is not explained (PE)

ESBL-PE is explained in Line 43. I hope you can check it out.

20 December 2023

Best regards.

Takanobu Hoshi

Reviewer 2 Report (New Reviewer)

Comments and Suggestions for Authors

Dear Authors,

The scope of the paper is very important and refers to the issue related to severe infections and their treatment. Escalation and de-escalation of antibiotic therapy is in some cases critical point of treatment and prevention against rapidly spreading antibiotic resistance among bacteria. Therefore, addressing its efficacy is of very high importance.

Introduction is well written and is finalized with the paragraph clearly stating the aim of the study.

Description of methods is also clear and there are no points as to which any critical remarks should be made.

The results obtained in this study were presented in the form of one figure (graph) and two tables. All is clear.

I only suggest to increase font in the graph, because some of its items are not very well visible.

One important remark:

There are numerous changes to the text marked in red, suggesting that this is some kind of corrected version of manuscript.

Author Response

Dear Reviewer 2,

I appreciate your valuable input.

Here is my response.

・I only suggest to increase font in the graph, because some of its items are not very well visible.

The font size of Figure 1 has been enlarged. Please check it.

・One important remark:

There are numerous changes to the text marked in red, suggesting that this is some kind of corrected version of manuscript.

This is a resubmitted manuscript and the Editor's instructions have marked the corrections in red.

20 December 2023

Best regards.

Takanobu Hoshi

Reviewer 3 Report (New Reviewer)

Comments and Suggestions for Authors

Dear Authors,

please find attached the suggestions and comments.

Author Response

Dear Reviewer 3,

I appreciate your valuable input.

Here is my response.

・Line 61: “using the disk method” - is the name of the method correct?

This is correct.

Many small and medium-sized hospitals in Japan use the disk diffusion method to confirm the determination of ESBL.

The description has been changed from "disk diffusion method" to "disk diffusion method".

(Line 65)

・Introduction: non-severe ESBL-EC bacteremia: describe the manifestations of the

disease in a few sentences; what is its importance; some data on the prevalence of

this disease.

I have cited and added the references you provided.

Lines 45-48- The incidence of Escherichia coli-producing ESBL (ESBL-EC) is increasing [4]. Individuals with ESBL-EC bloodstream infections (BSI) are reported to have a 14% longer hospital stay, a higher risk of recurrence [4], and higher mortality in critically ill patients in the intensive care unit [5] than those with non-ESBL-EC BSI.

・Material and methods: lack of reference to the literature.

References have been added to lines 109, 121 and 123.

Line 109- estimated glomerular filtration rate [15]

Line 121- Clinical and Laboratory Standards Institute (CLSI) M100-S26 on the Micros can Negative MIC panel for Enterobacterales 2J panel using the Micros can WalkAway 40  Plus [16].

Line 123- CLSI criteria [16].

・Lines: 132-137: Twenty-two patients were given an antimicrobial agent effective

against ESBL-EC within 3 days of the bacteremia and completed treatment with that

agent alone. Further, 14 and 8 patients were treated with CMZ and MEPM,

respectively. A total of 20 patients, including 9 treated with dCMZ and 11 with

eMEPM, were switched to other effective antimicrobial agents (de-escalation or

escalation) after starting an effective antimicrobial for ESBL-EC – the text is repeated

in lines 141-145.

Lines 141-145 were corrected.

Lines 146-148

Depending on whether the initial treatment was continued, eligible patients were divided into the CMZ group (n=14), MEPM group (n=8), dCMZ group (n=9), and eMEPM group (n=11).

・Line 156: For MIC distribution, there were no strains showing MICs other than 4 <8.

–please explain in more detail, highlight its importance.

Added explanation in lines 160-162.

Both <4 and <8 are susceptible  by CLSI criteria, but the percentage of <8 in the CMZ, MEPM, dCMZ, and eMEPM groups were 7.1%, 12.5%, 0%, and 36.4%, respectively.

・Literature research can also be improved, some suggestions:

Thank you for the literature suggestions.

I have cited the literature you suggested in the Introduction.

Lines 45-48

The incidence of Escherichia coli-producing ESBL (ESBL-EC) is increasing [4]. Individuals with ESBL-EC bloodstream infections (BSI) are reported to have a 14% longer hospital stay, a higher risk of recurrence [4], and higher mortality in critically ill patients in the intensive care unit [5] than those with non-ESBL-EC BSI.

Lines 52-53

Even the isolation of ESBL-EC from urine samples has been reported to affect the use of CBP [8].

20 December 2023

Best regards.

Takanobu Hoshi

Round 2

Reviewer 3 Report (New Reviewer)

Comments and Suggestions for Authors

The authors have responded to the comments and have also made corrections. Good luck for your further scientific work!

This manuscript is a resubmission of an earlier submission. The following is a list of the peer review reports and author responses from that submission.

Round 1

Reviewer 1 Report

Comments and Suggestions for Authors

Dear authors,

in your brief report you presented the results of the investigation related to the use of cefmetazole and meropenem in ESBL-EC patients. Manuscript is well written, Material and Methods are reproducible and could be repeated; Results are described both textually and graphically; Discussion mention other findings and compare it. Possible shortcomings are, as you also described,  low number of patients and lack of non-inferiority study. 

Specific comments:

title - please consider if wording "different administration methods" is right one for the title of this manuscript? did you use different administration method or different drug in this investigation? or different administration protocol? (In L 178 you mention "different approaches to antimicrobial therapy")

L 57- please mention reference for this statement ("has recently been introduced")

L 104-105 dots in these sentences

L 107 twice positive

L 116-117 and 127 please rephrase "patient who visited during the study period"

Author Response

Please find the attachment for detail.

Reviewer 2 Report

Comments and Suggestions for Authors

The manuscript titled, "Effect of different administration methods of cefmetazole and meropenem on the time to defervescence in non-severe extended-spectrum β-lactamase-producing Escherichia coli bacteremia." present interesting data of the differences in clinical efficacy between early CMZ and meropenem (MEPM) administration in ESBL-EC bacteremia and the evaluation of the usefulness of CMZ administration. In addition, they compared the cases of de-escalation from broad-spectrum antimicrobial agents to CMZ and escalation from CMZ to MEPM. Although there are several reports about the CMZ efficacy, there have been no reports on using the time to defervescence to measure clinical efficacy in ESBL-EC bacteremia. The results indicated that early CMZ administration in patients with non-severe ESBL-EC bacteremia may result in a more prolonged fever resolution than MEPM. Although there are weaknesses in that the non-inferiority studies to evaluate the equivalence between MEPM and CMZ are lacking, and also that the number of patients in this single-center study was low, I believe the manuscript of interest to the reader of Infectious Disease Reports as a brief report.

Reviewer 3 Report

Comments and Suggestions for Authors

One of the most easy to read and evaluate article, this year.
In my opinion, it is worth to be published. I had one methodological question - however, the authors answered it in discussion chapter).

Reviewer 4 Report

Comments and Suggestions for Authors

Takanobu Hoshi et al in this study compared the effects of cefmetazole (CMZ) and meropenem (MEPM) on the time to resolve fever in patients with extended-spectrum β-lactamase (ESBL)-producing Escherichia coli (ESBL-EC) bacteremia, excluding the influence of prior antimicrobial agents. Four groups were examined: CMZ, MEPM, de-escalation to CMZ (dCMZ), and escalation to MEPM (eMEPM). The median time to fever resolution was 3.5, 1.0, 2.0, and 4.0 days in the CMZ, MEPM, dCMZ, and eMEPM groups, with no significant differences. However, the analysis showed that CMZ delayed fever resolution more than MEPM in patients with non-severe ESBL-EC bacteremia, with a significant hazard ratio of 0.378 (p = 0.046). This study looks very interesting, it is well written and i accept the manuscript in the present form